# LUBAC prevents lethal dermatitis by inhibiting cell death induced by TNF, TRAIL and CD95L

Lucia Taraborrelli[1], Nieves Peltzer[1], Antonella Montinaro[1], Sebastian Kupka[1], Eva Rieser[1], Torsten Hartwig[1], Aida Sarr[1], Maurice Darding[1], Peter Draber[1], Tobias L. Haas[2], Ayse Akarca[3], Teresa Marafioti[3], Manolis Pasparakis [4], John Bertin[5], Peter J. Gough [5], Philippe Bouillet[6,7], Andreas Strasser[6,7], Martin Leverkus[8], John Silke [6,7] & Henning Walczak[1]

The linear ubiquitin chain assembly complex (LUBAC), composed of HOIP, HOIL-1 and SHARPIN, is required for optimal TNF-mediated gene activation and to prevent cell death induced by TNF. Here, we demonstrate that keratinocyte-specific deletion of HOIP or HOIL-1 (E-KO) results in severe dermatitis causing postnatal lethality. We provide genetic and pharmacological evidence that the postnatal lethal dermatitis in $Hoip^{E-KO}$ and $Hoil-1^{E-KO}$ mice is caused by TNFR1-induced, caspase-8-mediated apoptosis that occurs independently of the kinase activity of RIPK1. In the absence of TNFR1, however, dermatitis develops in adulthood, triggered by RIPK1-kinase-activity-dependent apoptosis and necroptosis. Strikingly, TRAIL or CD95L can redundantly induce this disease-causing cell death, as combined loss of their respective receptors is required to prevent TNFR1-independent dermatitis. These findings may have implications for the treatment of patients with mutations that perturb linear ubiquitination and potentially also for patients with inflammation-associated disorders that are refractory to inhibition of TNF alone.

[1] Centre for Cell Death, Cancer, and Inflammation (CCCI), UCL Cancer Institute, University College London, 72 Huntley Street, London WC1E 6DD, UK. [2] Institute of General Pathology, Università Cattolica del Sacro Cuore, 00168 Rome, Italy. [3] Department of Cellular Pathology, University College London, 21 University Street, London WC1E 6DE, UK. [4] Cologne Excellence Cluster on Cellular Stress Responses in Aging-Associated Diseases (CECAD), and Center for Molecular Medicine (CMMC), University of Cologne, D-50931 Cologne, Germany. [5] Pattern Recognition Receptor Discovery Performance Unit, Immuno-Inflammation Therapeutic Area, GlaxoSmithKline, Collegeville, PA 19422, USA. [6] The Walter and Eliza Hall Institute of Medical Research, 1G Royal Parade, Parkville, VIC 3052, Australia. [7] The Department of Medical Biology, The University of Melbourne, Parkville, VIC 3050, Australia. [8] Department of Dermatology & Allergology, University Hospital of RWTH Aachen University, 52074 Aachen, Germany. These authors contributed equally: Lucia Taraborrelli, Nieves Peltzer. Deceased: Martin Leverkus. Correspondence and requests for materials should be addressed to H.W. (email: h.walczak@ucl.ac.uk)

A proper balance between cell death, proliferation and differentiation maintains homeostasis of the skin that is critical to retain this organ's vital role as an immunological barrier against pathogens and as a physical barrier to water loss and mechanical insults. If any of these processes are deregulated, pathological conditions, such as skin infections, autoinflammatory and autoimmune disorders or cancer can occur[1]. Members of the tumour necrosis factor (TNF) and TNF receptor (TNFR) superfamilies are essential mediators of cell death and inflammation and play critical roles in innate as well as adaptive immune responses in many tissues and cell types, including in the skin and epidermal keratinocytes[2].

Engagement of TNFR1 by TNF induces formation of the TNFR1 signalling complex (TNFR1-SC), also referred to as complex I of TNFR1 signalling, an event that triggers gene activation by nuclear factor (NF)-κB and transcription factors activated downstream of mitogen-activated protein kinases[3,4]. TNFR1 signalling can, however, also result in cell death. This is triggered by a secondary cytoplasmic complex, also called complex II, which is formed by the recruitment of Fas-associated protein with a death domain (FADD) and caspase-8 to receptor-interacting serine/threonine-protein kinase 1 (RIPK1). In this platform, caspase-8 is cleaved and thereby activated, inducing apoptotic cell death[5,6]. Alternatively, when caspase-8 is inhibited or either FADD or caspase-8 is absent, RIPK1 recruits RIPK3, which in turn activates mixed lineage kinase domain-like protein (MLKL), resulting in the induction of regulated necrosis, also referred to as necroptosis[7,8]. However, complex II formation and activity is minimised when complex I is properly assembled and activated[9–14].

The linear ubiquitin chain assembly complex (LUBAC) regulates the balance between gene activation and cell death upon engagement of TNFR1 and certain other innate and adaptive immune receptors including Toll-like receptors (TLRs), TNF-related apoptosis-inducing ligand (TRAIL), NOD-like receptors and T and B cell receptors[15–20]. LUBAC, composed of three proteins, Heme-oxidized IRP2 ubiquitin ligase 1 (HOIL-1), Shank-associated RH domain-interacting protein (SHARPIN) and HOIL-1-interacting protein (HOIP), is the only E3 ligase identified so far capable of generating linear ubiquitin linkages de novo[21–25]. We previously showed that LUBAC prevents complex II formation upon TNFR1 stimulation, thereby inhibiting TNFR1-mediated cell death[26–28]. Mice deficient for SHARPIN, known as chronic proliferative dermatitis mice (cpdm) and referred to as $Sharpin^{cpdm/cpdm}$ mice herein, suffer from severe inflammation in the skin and other organs[29–31], which is caused by excessive TNFR1-mediated death of keratinocytes[22,27,32]. In contrast, deficiency in HOIP or HOIL-1 results in embryonic lethality[26,28,33]. The differences in the phenotypes of mice deficient for the different LUBAC components is due to the fact that in the absence of HOIP or HOIL-1 there is a complete lack of linear ubiquitination in complex I, whereas in the absence of SHARPIN it is merely reduced[28]. Thus, whereas HOIP and HOIL-1 are both essential for LUBAC activity, SHARPIN only contributes to it.

To explore the role of HOIP and HOIL-1 in the control of epidermal cell death and skin homeostasis, we sought to investigate the effect of deleting them in keratinocytes. Surprisingly, we found that keratinocyte-specific deletion of HOIP or HOIL-1 results in a lethal inflammatory skin disease, which is only partially dependent on TNFR1-induced cell death. The TNFR1-independent dermatitis is also a consequence of cell death that can, intriguingly, be redundantly triggered by TRAIL or CD95L. These findings identify a vital and previously unrecognised physiological role of HOIP and HOIL-1 in preventing cell death-induced inflammation, importantly beyond TNF as the only endogenous inducer of this cell death.

## Results

### HOIP and HOIL-1 are essential to maintain skin homeostasis.
To understand the role of LUBAC in the skin, we generated mice that lack HOIP or HOIL-1 selectively in epidermal keratinocytes by crossing *Hoip*- and *Hoil-1*-floxed mice with mice expressing the Cre recombinase under the control of the human keratin 14 (K14) promoter. The genotype of the mice was confirmed by PCR (Supplementary Fig. 1a). At the protein level, deletion of HOIP or HOIL-1 in keratinocytes was verified by western blot and immunohistochemistry (Supplementary Fig. 1b, c). As expected, HOIP deficiency abrogated linear ubiquitination at the TNFR1-SC (Supplementary Fig. 1d) and reduced TNFR1-mediated NF-κB activation in primary murine keratinocytes (PMKs) without preventing it (Supplementary Fig. 1e). Mice homozygous for keratinocyte-specific deletion of HOIP or HOIL-1 ($Hoip^{E-KO}$ and $Hoil-1^{E-KO}$ mice, respectively) were born at the expected Mendelian frequencies and were macroscopically indistinguishable from littermates up to postnatal day (P) 2 (data not shown). From this day onwards, however, both $Hoip^{E-KO}$ and $Hoil-1^{E-KO}$ mice developed severely damaged and scaly skin, which, invariably, resulted in the death of these mice between P4 and P6 (Fig. 1a). No *Hoip* or *Hoil-1* gene dosage effect was observed as $Hoip^{fl/wt}K14Cre^+$ and $Hoil-1^{fl/wt}K14Cre^+$ mice developed normally into adulthood without showing any signs of skin disease (data not shown).

Histological analysis of $Hoip^{E-KO}$ and $Hoil-1^{E-KO}$ mice at P4 revealed increased epidermal thickness, parakeratosis, hyperkeratosis and keratinocyte differentiation defects (Fig. 1b, c). These pathologies were accompanied by abnormal myeloid cell infiltration and high levels of cell death as demonstrated by increased cleaved caspase-3 and terminal deoxinucleotidyl transferase-mediated dUTP-fluorescein nick end labelling (TUNEL) staining (Fig. 1b, d, e and Supplementary Fig. 1f, g). Together, these observations reveal that HOIP and HOIL-1 are essential to prevent fatal dermatitis characterised by disruption of the normal epidermal structure, inflammation and aberrant keratinocyte death.

### Lethal dermatitis is only partially mediated by TNFR1.
The inflammatory syndrome of $Sharpin^{cpdm/cpdm}$ mice can be abrogated by the absence of TNF and TNFR1 or by the loss of the kinase activity of RIPK1[22,27,32,34]. We therefore first tested whether genetic ablation of TNFR1 could also prevent the morbidity and mortality in $Hoip^{E-KO}$ and $Hoil-1^{E-KO}$ mice. Unexpectedly, however, inflammation was only delayed in $Tnfr1^{KO};Hoip^{E-KO}$ and $Tnfr1^{KO};Hoil-1^{E-KO}$ mice as they progressively developed severe skin lesions resulting in a median survival of 70 days (Fig. 2a and Supplementary Fig. 2a). Sick $Tnfr1^{KO};Hoip^{E-KO}$ and $Tnfr1^{KO};Hoil-1^{E-KO}$ mice presented with epidermal disruption and thickening, parakeratosis, hyperkeratosis and inflammation (Fig. 2b, c). Crucially, infiltration by myeloid and lymphoid cells and cell death were significantly augmented in the epidermis of adult $Tnfr1^{KO};Hoip^{E-KO}$ mice compared to control animals (Fig. 2b, d and Supplementary Fig. 2b–d).

Next, we addressed whether ablation of the kinase activity of RIPK1 was sufficient to prevent inflammation in $Hoip^{E-KO}$ mice. Surprisingly, genetic ablation of the kinase activity of RIPK1 was substantially less effective than loss of TNFR1 in preventing dermatitis as $Ripk1^{D138N};Hoip^{E-KO}$ mice died at around P8 from severe skin disease (Fig. 2f, g). Thus lethal dermatitis caused by keratinocyte-specific deficiency in either HOIP or HOIL-1, the

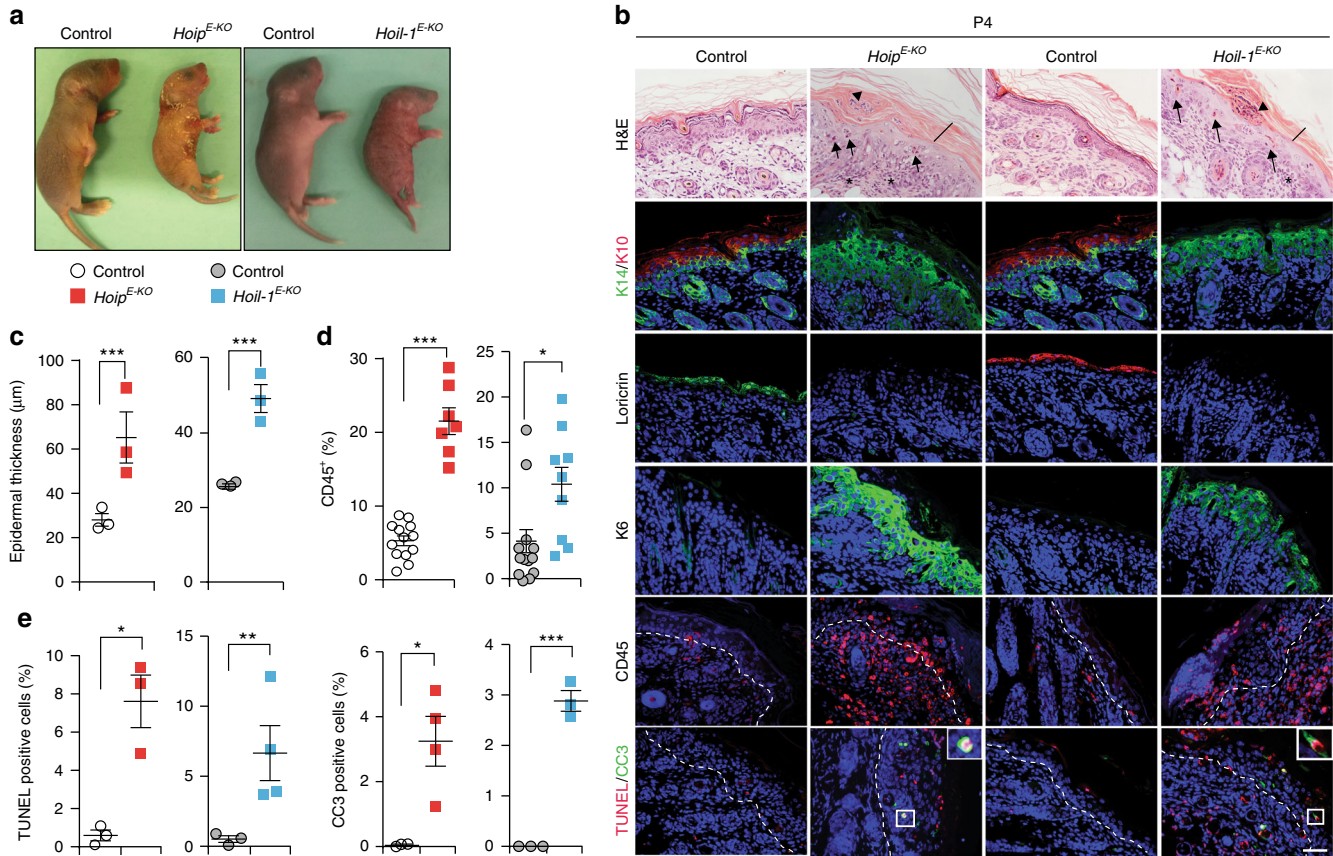

**Fig. 1** Deletion of HOIP or HOIL-1 in keratinocytes results in lethal dermatitis. **a** Representative images of mice of the indicated genotypes ($n = 10$ mice per genotype) at P4. **b** Representative images of skin sections stained with H&E, the indicated antibodies or TUNEL from mice of the indicated genotypes ($n = 3$ mice per genotype). Arrows: pyknotic nuclei, stars: immune cell infiltrates, arrowhead: parakeratosis and black bar: hyperkeratosis. Nuclei were stained with DAPI (blue). White dashed lines indicate boundary of epidermis (above) and dermis (below). Scale bar, 50 μm. **c** Epidermal thickness quantification at P4. Data are presented as mean values ± s.e.m. ($n = 3$ per genotype). ***$P \leq 0.001$. **d** Percentages of CD45+ cells in the skin of the indicated genotypes at P4. Data are presented as mean values ± s.e.m. ($n < 8$ per genotype), *$P \leq 0.05$, ***$P \leq 0.001$. **e** Quantification of TUNEL- and CC3-positive cells as represented in **b**. Data are presented as mean values ± s.e.m. ($n = 3$ mice per genotype), *$P \leq 0.05$, **$P \leq 0.01$, ***$P \leq 0.001$. Control mice represent a pool of *Hoip*$^{fl/fl}$;*K14-Cre−* and *Hoip*$^{fl/wt}$;*K14-Cre+* (white circles) or *Hoil-1*$^{fl/fl}$;*K14-Cre−* and *Hoil-1*$^{fl/wt}$;*K14-Cre+* (grey circles). P: postnatal day

two essential components of LUBAC[28], is only partially dependent on TNFR1 and, in the presence of TNFR1, almost completely independent of the kinase activity of RIPK1.

**Increased cell death precedes inflammation.** We next investigated the temporal relationship between aberrant cell death and inflammation in *Hoip*$^{E-KO}$ and *Hoil-1*$^{E-KO}$ mice. Abnormally increased cell death in the epidermis of *Hoip*$^{E-KO}$ and *Hoil-1*$^{E-KO}$ mice was already apparent in utero at E18.5 and at birth (P0) (Fig. 3a, b and Supplementary Fig. 3a, b). This implies that lack of linear ubiquitination in keratinocytes results in aberrant cell death in sterile conditions. *Hoip*$^{E-KO}$ and *Hoil-1*$^{E-KO}$ mice displayed abnormally increased immune cell infiltration at P2 but not at birth (Fig. 3a, c and Supplementary Fig. 3c–e). Accordingly, keratinocyte differentiation and epidermal thickness appeared abnormal at P2 but not at E18.5 or P0 (Fig. 3a and Supplementary Fig. 3f). Thus excessive cell death precedes the inflammatory response suggesting that keratinocyte death upon loss of HOIP or HOIL-1 may trigger lethal dermatitis.

To understand the mechanism of cell death induction in the skin of *Hoip*$^{E-KO}$ and *Hoil-1*$^{E-KO}$ mice, we analysed the formation of the signalling platform known to trigger cell death downstream of various death receptors[35] by immunoprecipitating the adaptor protein FADD in PMKs in the presence of caspase inhibitor,

Z-VAD-fmk. This revealed that, even without an exogenous stimulus, a FADD/caspase-8/RIPK1-containing complex was readily detectable in HOIP-deficient but not in control PMKs (Fig. 4a). Consistent with apoptotic signalling by such a complex, the HOIP-deficient cells were less viable even in the absence of exogenous stimuli (Fig. 4b). This loss in cell viability was significantly reduced by inhibition of caspases and RIPK1 kinase activity but not by blocking the kinase activity of RIPK3 (Fig. 4b). Inhibition of TNF or genetic ablation of TNFR1 also increased the viability of *Hoip*$^{E-KO}$ PMKs (Fig. 4c, d). These results indicate that in PMKs HOIP prevents aberrant RIPK1 kinase-dependent apoptosis triggered by spontaneously produced autocrine TNF.

**LUBAC loss in adulthood induces cell death-driven dermatitis.** To assess the impact of acute loss of HOIP in keratinocytes in adult mice, we treated *Hoip*$^{fl/fl}$*K14CreER*$^{Tam}$ mice with 4-hydroxytamoxifen (4-OHT) in a localised area of the skin. This treatment resulted in rapid cell death induction, followed by increased immune cell infiltration (Fig. 5a–c and Supplementary Fig. 4a). This was accompanied by epidermal thickening, hyperplasia, hyperkeratosis as well as parakeratosis and defects in keratinocyte differentiation (Fig. 5d and Supplementary Fig. 4b–e). These findings demonstrate that HOIP is required to maintain normal skin architecture and function in adult mice and that also

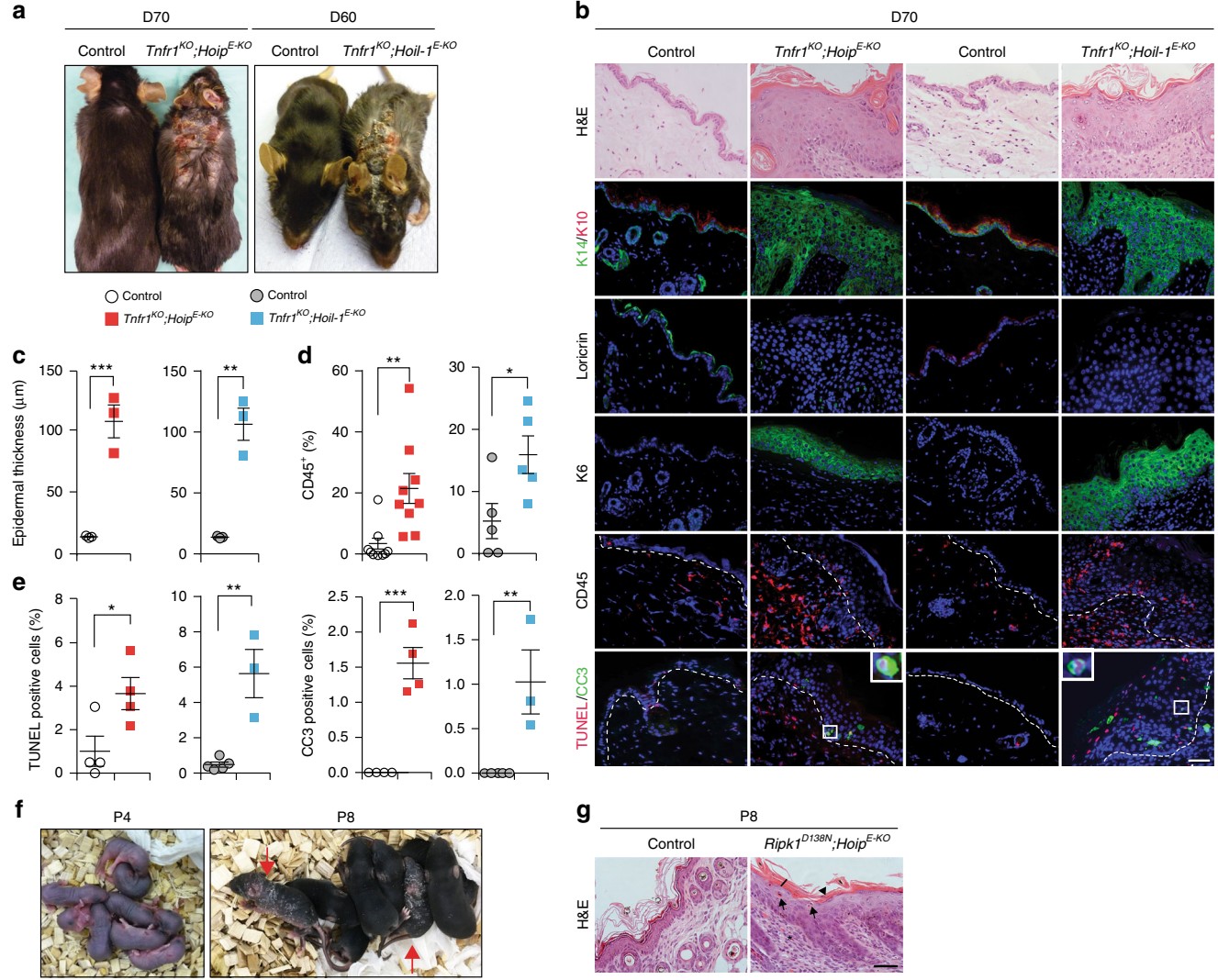

**Fig. 2** Dermatitis of *Hoip*[E-KO] and *Hoil-1*[E-KO] mice is delayed by TNFR1 deficiency but not by abrogation of the kinase activity of RIPK1. **a**, **f** Representative images of mice of the indicated genotypes, (*n* = 10 mice per genotype). Arrows in **f** indicate *Ripk1*[D138N];*Hoip*[E-KO] mice at P8 (right panel). *Ripk1*[D138N]; *Hoip*[E-KO] mice were indistinguishable from control littermates at P4 (left panel) (**f**). **b** Representative images of skin sections stained with H&E, the indicated antibodies or TUNEL from mice of the indicated genotypes (*n* = 3 mice per genotype). Nuclei were stained with DAPI (blue). White dashed lines indicate boundary of epidermis (above) and dermis (below). Scale bar, 50 μm. **c** Epidermal thickness quantification at D70. Data are presented as mean values ± s.e.m. (*n* = 3 per genotype). **P ≤ 0.01, ***P ≤ 0.001. **d** Percentages of CD45[+] cells in the skin of the indicated genotypes at D70. Data are presented as mean values ± s.e.m. (*n* < 5 per genotype), *P ≤ 0.05, **P ≤ 0.01. **e** Quantification of TUNEL- and CC3-positive cells as represented in **b**. Data are presented as mean values ± s.e.m. (*n* = 3 mice per genotype). *P ≤ 0.05, **P ≤ 0.01, ***P ≤ 0.001. **g** Representative images of skin sections stained with H&E from mice of the indicated genotypes (*n* = 3 mice per genotype). Scale bar, 50 μm. Arrows: pyknotic nuclei, stars: immune cell infiltrates, arrowhead: parakeratosis and black bar: hyperkeratosis. Control mice represent a pool of *Tnfr1*[KO];*Hoip*[fl/fl];*K14-Cre−* and *Tnfr1*[KO];*Hoip*[fl/wt];*K14-Cre+* mice (white circles) or *Tnfr1*[KO];*Hoil-1*[fl/fl];*K14-Cre−* and *Tnfr1*[KO];*Hoil-1*[fl/wt];*K14-Cre+* mice (grey circles) (**c**) and *Ripk1*[D138N];*Hoip*[fl/fl];*K14-Cre−* and *Ripk1*[D138N];*Hoip*[fl/wt];*K14-Cre+* mice (**g**). P: postnatal day, D: day

in the adult skin cell death precedes inflammation in the absence of HOIP.

**Aberrant apoptosis is responsible for lethal dermatitis**. Since aberrantly increased cell death was the first abnormal event we could detect in the epidermis of *Hoip*[E-KO] and *Hoil-1*[E-KO] mice and because it is also observed in the absence of TNFR1 in adult mice, we next evaluated genetically whether and, if so, which form(s) of aberrant cell death cause the early and the late dermatitis.

Consistent with the apoptotic cell death observed in vitro, genetic ablation of *Ripk3* in *Hoil-1*[E-KO] or the loss of *Mlkl* in *Hoip*[E-KO] mice failed to prevent aberrant cell death and skin inflammation and did not delay the postnatal lethality (Fig. 6b, c, e and Supplementary Fig. 5).

Since caspase-8 deficiency is embryonically lethal due to sensitisation to RIPK3- and MLKL-induced necroptosis[9,10,36–39], it is not possible to generate viable *Casp8*[KO];*Hoil-1*[E-KO] mice. We therefore first evaluated the effect of *Casp8* heterozygosity in *Hoil-1*[E-KO] and *Ripk3*[KO];*Hoil-1*[E-KO] mice. Heterozygosity of *Casp8* was able to extend the survival of *Hoil-1*[E-KO] and *Ripk3*[KO];*Hoil-1*[E-KO]

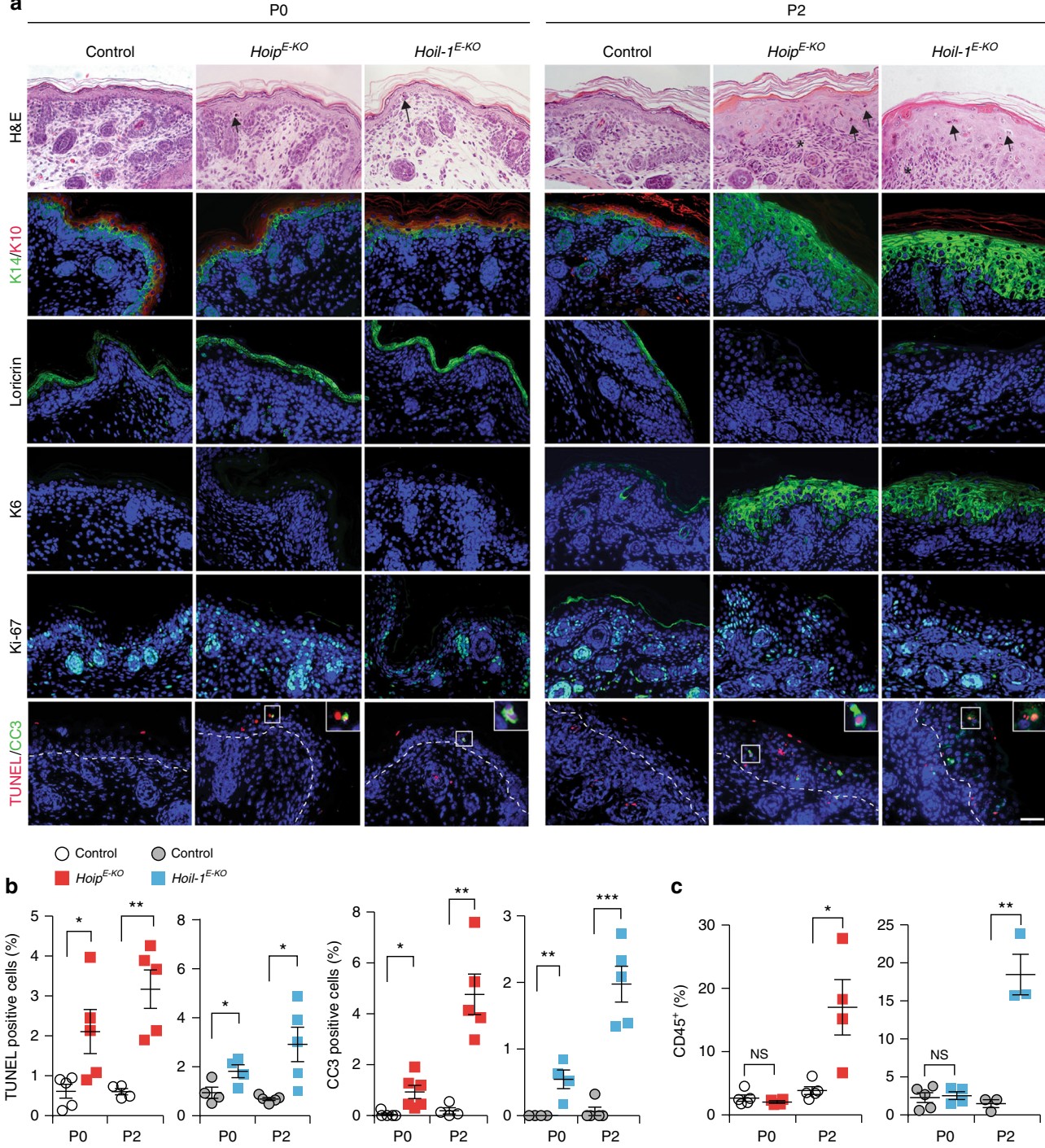

**Fig. 3** Cell death precedes the onset of inflammation in *Hoip*[E-KO] and *Hoil-1*[E-KO] mice. **a** Representative images of skin sections stained with H&E, the indicated antibodies or TUNEL from mice of the indicated genotypes ($n = 3$ mice per genotype). Arrows: pyknotic nuclei, stars: immune cell infiltrates. Nuclei were stained with DAPI (blue). White dashed lines indicate boundary of epidermis (above) and dermis (below). Scale bar, 50 μm. **b** Quantification of TUNEL- and cleaved caspase-3 (CC3)-positive cells in skin sections from mice of the indicated genotypes. Data are presented as mean values ± s.e.m. ($n < 4$ mice per genotype). *$P \leq 0.05$, **$P \leq 0.01$, ***$P \leq 0.001$. **c** Percentages of CD45[+] cells in the skin of the indicated genotypes at the indicated stages. Data are presented as mean values ± s.e.m. ($n < 3$ per genotype). **$P \leq 0.01$, ***$P \leq 0.001$, NS: not significant. Control mice represent a pool of *Hoip*[fl/fl]; *K14-Cre−* and *Hoip*[fl/wt];*K14-Cre+* (white circles) or *Hoil-1*[fl/fl];*K14-Cre−* and *Hoil-1*[fl/wt];*K14-Cre+* (grey circles). P: postnatal day

mice to around P8 and day 20, respectively (Fig. 6e and Supplementary Fig. 6a–d).

We next determined the effect of complete absence of caspase-8. Remarkably, both *Mlkl*[KO];*Casp8*[KO];*Hoip*[E-KO] and *Ripk3*[KO];*Casp8*[KO]; *Hoil-1*[E-KO] mice reached adulthood without any signs of skin disease

(Fig. 6a and Supplementary Fig. 6e, f). Epidermal structure and keratinocyte differentiation were completely normal in *Ripk3*[KO]; *Casp8*[KO];*Hoil-1*[E-KO] mice, and these animals neither exhibited aberrant cell death nor immune cell infiltration in their skin (Fig. 6b–d and Supplementary Fig. 6g). *Ripk3*[KO];*Casp8*[KO];*Hoil-1*[E-KO]

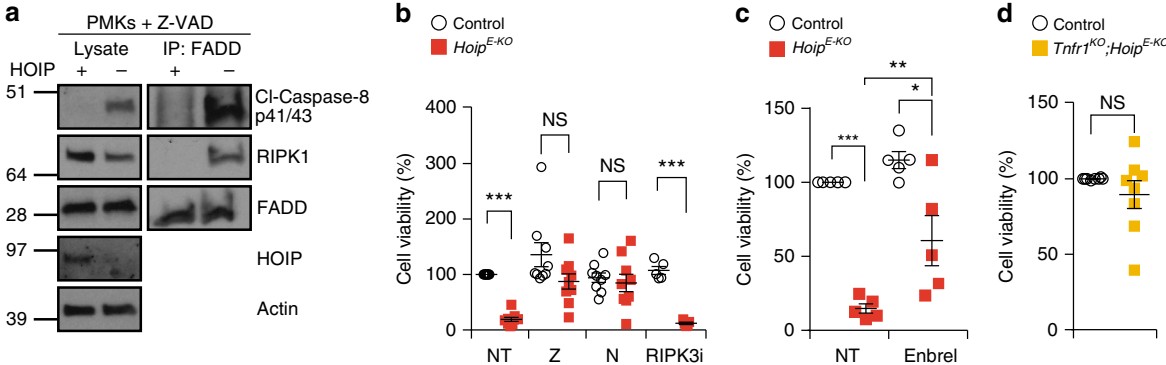

**Fig. 4** HOIP-deficient PMKs display TNF-mediated RIPK1 kinase-dependent apoptosis. **a** FADD-IP was performed in primary mouse keratinocytes (PMKs) derived from control (+) or $Hoip^{E-KO}$ (−) mice cultured in the presence of the caspase inhibitor Z-VAD-fmk (n = 2 independent experiment). Lysates and IP were analysed by western blotting for the indicated proteins. See Supplementary Fig. 10 for source data blots. **b** Percentage of cell viability in PMKs derived from $Hoip^{E-KO}$ and control mice were cultured for 4 days in the absence (NT: not-treated) or presence of the inhibitors Necrostatin-1s (N), Z-VAD-fmk (Z) or RIPK3i. Data are presented as mean values ± s.e.m. (n < 5 per genotype). ***$P \leq 0.001$, NS: not significant. **c** Cell viability (%) of PMKs derived from $Hoip^{E-KO}$ and control mice were cultured with or without (NT) Etanercept (Enbrel®). Results are presented as mean values ± s.e.m. (n = 5 mice per genotype). *$P \leq 0.05$, **$P \leq 0.01$, ***$P \leq 0.001$. **d** Cell viability (%) of PMKs derived from adult mice with the indicated genotypes. Results are presented as mean values ± s.e.m. (n = 8 mice per genotype). NS: not significant. Control mice represent a pool of $Hoip^{fl/fl};K14\text{-}Cre-$ and $Hoip^{fl/wt};K14\text{-}Cre+$ (**b**, **c**) and $Tnfr1^{KO};Hoip^{fl/fl};K14\text{-}Cre-$ and $Tnfr1^{KO};Hoip^{fl/wt};K14\text{-}Cre+$ mice (**d**)

mice survived without developing any signs of skin inflammation well beyond the 70-day time point at which the $Tnfr1^{KO};Hoil\text{-}1^{E-KO}$ mice succumb to severe dermatitis (Fig. 6e), although they had to be sacrificed at later times because of lymphadenopathy and splenomegaly (Supplementary Fig. 6h) caused by the combined deficiency in caspase-8 and RIPK3 or MLKL, as previously reported[10,38,40]. Collectively, these results demonstrate that caspase-8-mediated apoptosis is responsible for the lethal dermatitis in mice lacking HOIP or HOIL-1 in keratinocytes and that RIPK3/MLKL-mediated necroptosis does not contribute to the disease.

**RIPK1 kinase is required for TNFR1-independent dermatitis.**
Since aberrant cell death is the cause of dermatitis in $Tnfr1^{KO};$ $Hoip^{E-KO}$ and $Tnfr1^{KO};Hoil\text{-}1^{E-KO}$ mice, we aimed to further characterise this type of cell death. In order to evaluate the contribution of necroptosis to the pathology of $Tnfr1^{KO};Hoip^{E-KO}$ mice, we generated $Mlkl^{KO};Tnfr1^{KO};Hoip^{E-KO}$ mice. These mice developed less severe skin lesions at day 70 and their survival was significantly prolonged (Fig. 7a–c). Thus, in the absence of TNFR1, necroptosis contributes to skin inflammation.

To investigate the involvement of the kinase activity of RIPK1 in the TNFR1-independent disease caused by LUBAC deficiency in the skin, we next fed $Tnfr1^{KO};Hoil\text{-}1^{E-KO}$ and control mice from E14.5 until day 100 with the RIPK1 inhibitor GSK'547A[41]. As a control, $Sharpin^{cpdm/cpdm}$ and $Hoip^{E-KO}$ mice were also fed with GSK'547A. In line with the genetic analysis of $Hoip^{E-KO}$ mice (Fig. 2f, g) and previous reports with $Sharpin^{cpdm/cpdm}$ mice[34], this treatment delayed disease development and death of $Hoip^{E-KO}$ mice up to P8 and prevented dermatitis in $Sharpin^{cpdm/cpdm}$ mice (Supplementary Fig. 7a, b). Strikingly, pharmacologic inhibition of the kinase activity of RIPK1 rescued the majority of $Tnfr1^{KO};Hoil\text{-}1^{E-KO}$ mice from dermatitis for the duration of the treatment with only three of the ten treated mice developing small punctate scales (Fig. 7c–e and Supplementary Fig. 7c). Thus the lethal dermatitis caused by keratinocyte-specific HOIL-1 that occurs in the absence of TNFR1 is mediated by RIPK1 kinase-dependent apoptosis and necroptosis.

**TNF, TRAIL and CD95L drive cell death and dermatitis.**
Finally, we sought to identify the instigator(s) of the TNFR1-independent cell death that is responsible for the fatal dermatitis

in $Hoip^{E-KO}$ and $Hoil\text{-}1^{E-KO}$ mice. Consistent with our previous findings in other cell types[17,42], PMKs derived from $Tnfr1^{KO};$ $Hoil\text{-}1^{E-KO}$ mice were more sensitive than control cells to induction of cell death by TRAIL, CD95 (Fas/APO-1) ligand (CD95L) or Polyinosinic:polycytidylic acid (Poly(I:C)) (Fig. 8a). We, therefore genetically ablated the death domain (DD) of CD95 specifically in keratinocytes or deleted TRAIL-R or TLR3 systemically in $Tnfr1^{KO};Hoil\text{-}1^{E-KO}$ mice. However, $Cd95^{E-}$ $^{DD};Tnfr1^{KO};Hoil\text{-}1^{E-KO}$, $Trail\text{-}r^{KO};Tnfr1^{KO};Hoil\text{-}1^{E-KO}$ and $Tlr3^{KO};Tnfr1^{KO};Hoil\text{-}1^{E-KO}$ mice all suffered from skin lesions that were indistinguishable in severity from those seen in the $Tnfr1^{KO};Hoil\text{-}1^{E-KO}$ mice (Supplementary Fig. 8).

Reasoning that these death receptors might be able to drive disease in a redundant manner in $Tnfr1^{KO};Hoil\text{-}1^{E-KO}$ mice, we next assessed the impact of their combined loss on the onset of dermatitis. Co-deletion of TRAIL-R and TLR3 in $Tnfr1^{KO};Hoil\text{-}$ $1^{E-KO}$ mice resulted in slightly milder skin lesions at day 70 (Fig. 8b, c), yet these mice still succumbed to inflammatory skin disease with a median survival of 77 days (Fig. 8d). Strikingly, however, combined loss of TRAIL-R with keratinocyte-specific deletion of the DD of CD95 resulted in prevention of dermatitis in $Tnfr1^{KO};Hoil\text{-}1^{E-KO}$ mice reflected by the significantly reduced severity score of skin lesions and prolonged survival (Fig. 8b–d). We therefore conclude that the cell death-driven inflammatory disease caused by HOIL-1 deficiency in keratinocytes is induced by multiple death receptor-ligand systems, namely the TNF/ TNFR1, TRAIL/TRAIL-R and CD95L/CD95 systems. Thus, cell death induction via either of these receptors is sufficient to cause disease (Supplementary Fig. 9) in the absence of linear ubiquitination caused by HOIL-1 deficiency.

## Discussion
Our study shows that both HOIP and HOIL-1 are essential to maintain skin homeostasis by preventing skin inflammation caused by death receptor-induced cell death. Intriguingly, the dermatitis in $Hoip^{E-KO}$ and $Hoil\text{-}1^{E-KO}$ mice is different from that in $Sharpin^{cpdm/cpdm}$ mice, both with regards to severity and the mechanisms responsible for driving it. Whereas the dermatitis in $Hoip^{E-KO}$ and $Hoil\text{-}1^{E-KO}$ mice is driven by cell death induced by mechanisms beyond TNF/TNFR1, deletion of one copy of the $Tnf$ gene completely prevented dermatitis in $Sharpin^{cpdm/cpdm}$ mice[22].

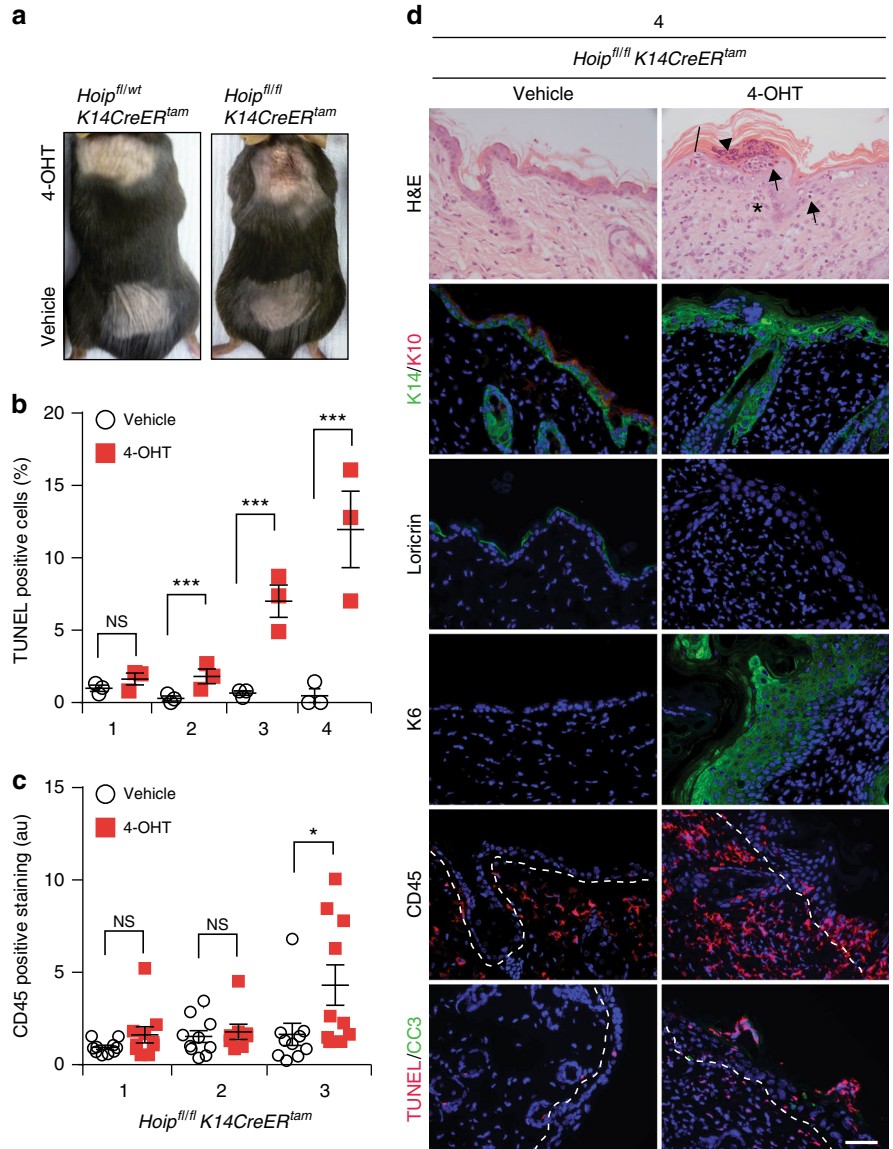

**Fig. 5** Adult *Hoip^(E-KO)* mice suffer from cell death-induced dermatitis. **a** Representative images of mice of the indicated genotypes (n = 8 mice per genotype). Animals were treated with vehicle or 4-OHT every other day for a total of four doses. **b** Quantification of TUNEL-positive cells in skin sections of *Hoip^(fl/fl)K14CreER^(tam)* mice after one, two, three or four treatments with vehicle or 4-OHT. Data are presented as mean values ± s.e.m. (n = 3 per genotype). ***P ≤ 0.001, NS: not significant. **c** Quantification of CD45 staining in skin sections from *Hoip^(fl/fl)K14CreER^(tam)* mice treated as in **b** was performed by measuring overall fluorescence intensity using ImageJ. Results are presented as mean values ± s.e.m. (n = 3 mice per genotype). *P ≤ 0.05, NS: not significant. au: arbitrary units. **d** Representative images of skin sections stained with H&E, the indicated antibodies and TUNEL from mice of the indicated genotypes (n = 3 mice per genotype) after four 4-OHT treatments. Arrows: pyknotic nuclei, stars: immune cell infiltrates, arrowhead: parakeratosis and black bar: hyperkeratosis. Nuclei were stained with DAPI (blue). White dashed lines indicate boundary of epidermis (above) and dermis (below). Scale bars, 50 μm

This uncovers a physiological role for HOIP and HOIL-1, which is more complex regarding LUBAC function than the one we and others previously identified for SHARPIN[22,27,32]. Crucially, this additional complexity relies on cell death-inducing systems other than the TNF/TNFR1 system.

Curiously, while inhibition of the kinase activity of RIPK1 restores viability of HOIP-deficient PMKs in vitro, neither pharmacologic inhibition nor genetic impairment of RIPK1's kinase activity in *Hoip^(E-KO)* mice prevented keratinocyte death and consequent dermatitis. It therefore appears that the regulation of cell death in vivo is more complex than revealed by the study of PMKs ex vivo. It is interesting to note in this context that apoptosis dependent on the kinase activity of RIPK1 was also described to occur upon inhibition or absence of TAK1, the NF-κB essential modulator (NEMO) or IKKα/IKKβ following TNF stimulation[11,43,44].

We and others previously showed that TNF-induced cell death can cause inflammation and inflammation-associated diseases[22,45]. The results we present here provide additional evidence in support of cell death as an aetiology of inflammation-associated diseases. Importantly, however, we extend this concept to endogenous factors capable of inducing inflammatory cell death beyond TNF. Crucially, we discover that these factors can act in concert with TNF to induce inflammatory cell death. This means that patients with a disease aetiology similar to the one we report here may benefit from a

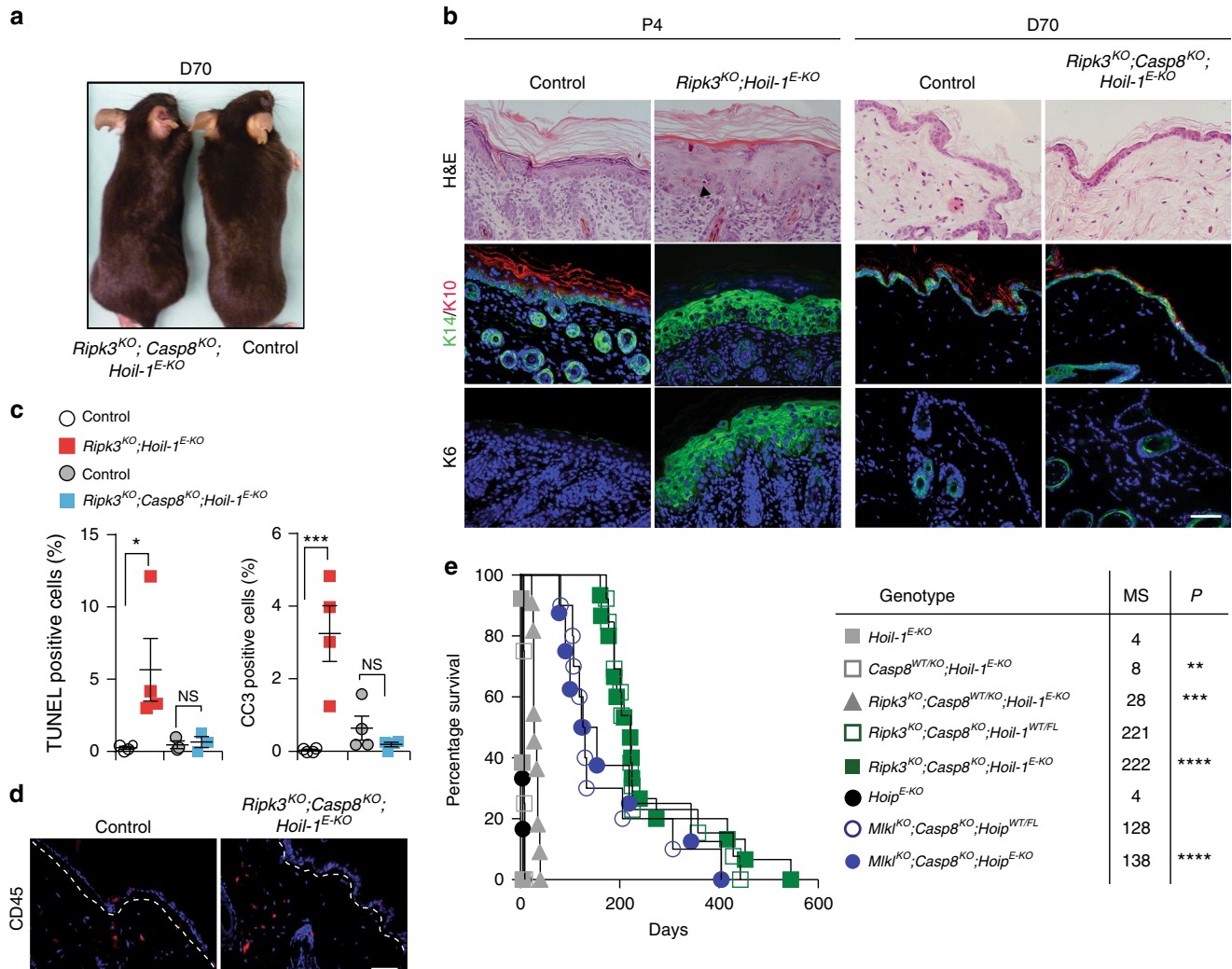

**Fig. 6** Aberrant apoptosis drives lethal dermatitis in *Hoip^E-KO* and *Hoil-1^E-KO* mice. **a** Representative images of mice of the indicated genotypes, (*n* = 15 mice per genotype). **b** Representative images of skin sections stained with H&E or with the indicated antibodies in mice with the indicated genotypes (*n* = 3 mice per genotype). Arrowhead: pyknotic nucleus. Nuclei were stained with DAPI (blue). Scale bars, 50 μm. **c** Quantification of TUNEL- and cleaved caspase-3 (CC3)-positive cells in skin sections from mice of the indicated genotypes. Data are presented as mean values ± s.e.m. (*n* = 4 mice per genotype). *P ≤ 0.05, ***P ≤ 0.001, NS: not significant. **d** Representative images of skin sections from mice of the indicated genotypes (*n* = 4 mice per genotype) stained with antibody against CD45 (red) at D70. Nuclei were stained with DAPI (blue). White dashed lines indicate boundary of epidermis (above) and dermis (below). Scale bar, 50 μm. **e** Kaplan–Meier survival curve of mice with the indicated genotypes. Comparisons between *Hoip^E-KO* (*n* = 10) and *Mlkl^KO*; *Casp8^KO*;*Hoip^E-KO* (*n* = 4) and, *Hoil-1^E-KO* (*n* = 13) and *Casp8^KO/WT*;*Hoil-1^E-KO* (*n* = 4), *Ripk3^KO*;*Casp8^KO/WT*;*Hoil-1^E-KO* (*n* = 11) or *Ripk3^KO*;*Casp8^KO*;*Hoil-1^E-KO* (*n* = 15) mice were submitted for statistical analysis. MS: median survival, **P ≤ 0.01, ***P ≤ 0.001, ****P ≤ 0.0001. *Ripk3^KO*;*Casp8^KO*;*Hoil-1^fl/wt*K14cre+ (*n* = 13) and *Mlkl^KO*;*Casp8^KO*;*Hoip^fl/wt*K14cre+ (*n* = 4) mice were used as controls. Control mice represent a pool of *Ripk3^KO*;*Hoil-1^fl/fl*;*K14-Cre−* and *Ripk3^KO*;*Hoil-1^fl/wt*;*K14-Cre+* or *Ripk3^KO*;*Casp8^KO*;*Hoil-1^fl/fl*;*K14-Cre−* and *Ripk3^KO*;*Casp8^KO*;*Hoil-1^fl/wt*;*K14-Cre+* mice (**a–d**). P: postnatal day, D: day

therapy that combines the inhibition of TNF, or TNF-induced death, with that of the kinase activity of RIPK1 and/or of other death-inducing cytokines, most importantly CD95L and TRAIL. It is noteworthy that patients with LUBAC-inactivating germline mutations, such as those with HOIL-1, HOIP and OTULIN mutations, suffer from immunodeficiency and autoinflammation[46–50]. Intriguingly, whereas TNF-inhibitory treatment only temporarily ameliorated the pathology in one of the HOIL-1-deficient patients who received it, the benefit to the OTULIN-deficient patients identified so far was substantial[47,49,50]. Our observation that TNF is not the sole driver of cell death-driven inflammation and the suggestion of potentially effective combinatorial therapeutic options, based on simultaneous genetic interference with different death receptor-ligand systems in our model of LUBAC deficiency in the skin, may have implications for the future treatment of patients harbouring mutations that perturb normal LUBAC activity.

TNF inhibition is effective in treating several autoinflammatory and autoimmune disorders, including rheumatoid arthritis, psoriasis and Crohn's disease[51–53]. However, a significant fraction of patients with these diseases fails to respond to anti-TNF treatment. It is tempting to speculate that autoimmune patients with a cell death aetiology of their disease may benefit from combining the inhibition of TNF with that of RIPK1 and/or CD95L and TRAIL. This concept could possibly extend to patients with a cell death-driven autoimmune or otherwise inflammation-associated disease in which TNF inhibition, at least when applied alone, has so far failed, including amyotrophic lateral sclerosis and multiple sclerosis[54,55].

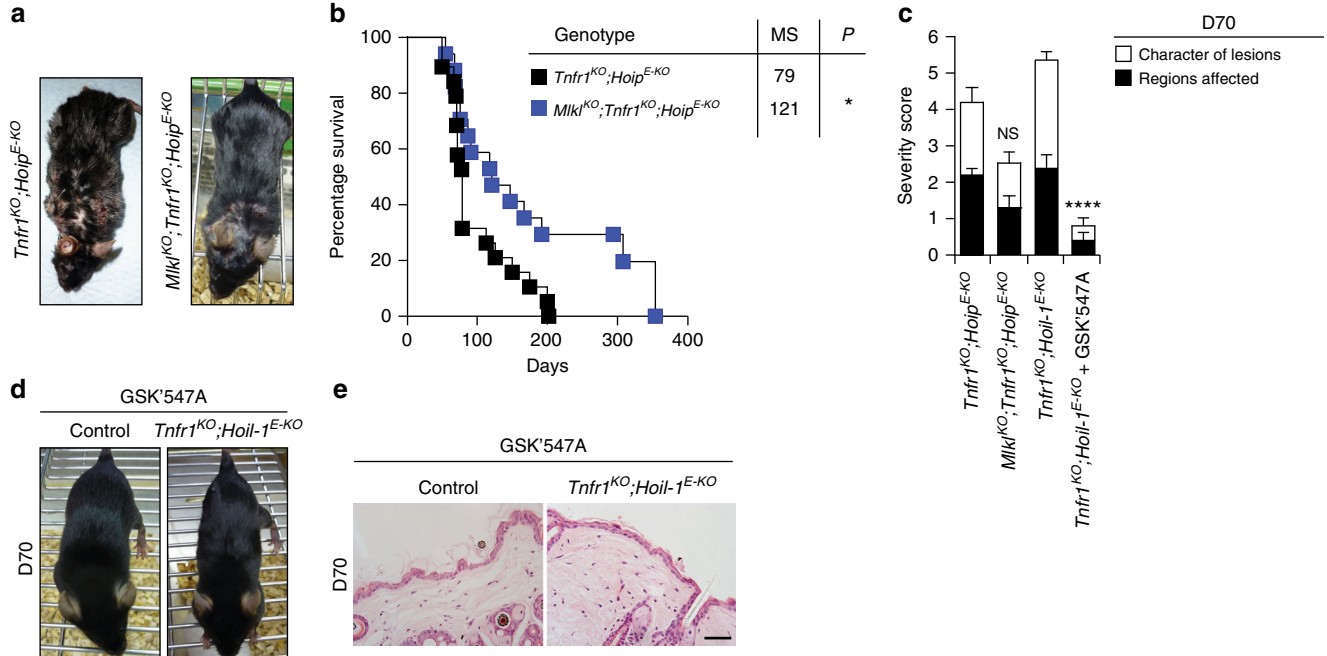

**Fig. 7** The RIPK1 kinase activity is required for TNFR1-independent dermatitis. **a**, **d** Representative images of mice of the indicated genotypes (**a**, **d**) and treatment (**d**). **b** Kaplan–Meier survival curve of mice with the indicated genotypes. Comparisons between $Tnfr1^{KO};Hoip^{E-KO}$ ($n = 14$) mice with $Mlkl^{KO};Tnfr1^{KO};Hoip^{E-KO}$ ($n = 17$) were submitted for statistical analysis. MS median survival *$P \leq 0.05$. **c** Severity score of dermatitis was assessed at D70 in mice of the indicated genotypes and treatment. $Tnfr1^{KO};Hoip^{E-KO}$ ($n = 5$), $Mlkl^{KO};Tnfr1^{KO};Hoip^{E-KO}$ ($n = 13$), $Tnfr1^{KO};Hoil-1^{E-KO}$ ($n = 6$) and $Tnfr1^{KO};Hoil-1^{E-KO}$+ GSK'547A ($n = 10$). Data are presented as mean values ± s.e.m. ****$P \leq 0.0001$, NS: not significant. **e** Representative images of skin sections stained with H&E from mice of the indicated genotypes fed with chow containing GSK'547A ($n = 3$ mice per genotype). Control mice represent a pool of $Tnfr1^{KO};Hoil-1^{fl/fl};$ $K14$-$Cre-$ and $Tnfr1^{KO};Hoil-1^{fl/wt};K14$-$Cre+$ mice (**d**, **e**)

## Methods

**Mice.** To generate $Hoip^{E-KO}$ and $Hoil-1^{E-KO}$ mice, $Hoip^{fl/fl[26]}$ and $Hoil-1^{fl/fl[28]}$ mice were crossed with mice expressing the Cre recombinase under the control of the human K14 promoter (obtained from Geert van Loo)[56], strain AZO-Nn4Cre (K14). The $K14CreER^{Tam[57]}$, $Mlkl^{KO[28]}$, $Ripk3^{KO[58]}$, $Casp8^{KO[59]}$, $Trail-r^{KO[60]}$ and $Ripk1^{D138N[61]}$ mice have been previously described. $Tnfr1^{KO}$, $Tlr3^{KO}$, $Cd95$-$DD^{fl/fl}$ mice (C57BL/6-Fastm1Cgn/J) and $Sharpin^{cpdm/cpdm}$ (C57BL/Ka) mice were purchased from The Jackson Laboratories. To induce deletion of HOIP in epidermal keratinocytes of adult mice, $Hoip^{fl/fl}K14CreER^{Tam}$ mice were treated as previously described[62]. Briefly, a small shaved area of the dorsal neck was treated with 50 μL of 4-OHT 20 mg mL$^{-1}$ dissolved in ethanol every other day for a total of 1, 2, 3 or 4 treatments, as indicated. As a vehicle treatment, a small dorsal area close to the tail was shaved and treated with ethanol. $Hoip^{fl/wt}K14CreER^{Tam}$ mice were used as tamoxifen controls. Mice were analyzed 2 days after the last treatment or as indicated in the figure legends. Timed matings were performed as previously described[26]. All mice were genotyped by PCR analysis. Colonies were fed ad libitum. All animal experiments were conducted under an appropriate UK project license in accordance with the regulations of UK home office for animal welfare according to ASPA (animal (scientific procedure) Act 1986).

**Pharmacological inhibition of RIPK1 kinase activity.** Pregnant females were fed with rodent chow containing 100 mg kg$^{-1}$ day$^{-1}$ GSK3540547A (GSK'547A) (GlaxoSmithKline LLC) from 14 days post coitum and continued the special diet throughout the nursing period. At weaning age, $Sharpin^{cpdm/cpdm}$ and $Tnfr1^{KO};$ $Hoil-1^{E-KO}$ mice and littermate controls were continuously treated with GSK'547A for another 100 days.

**Immunostaining and quantification.** Four-μm-thick formalin-fixed paraffin-embedded skin sections were stained following standard protocols. Briefly, sections were boiled in 10 mM sodium citrate buffer (pH 6.0) in a microwave. Slides were blocked in buffer containing Tween 20 0.5% and bovine serum albumin 0.2%. For CD45 staining, slides were boiled in Retrievagen A (BD) and blocked with buffer without Tween. Next, slides were incubated with the primary antibody overnight at 4 °C. The following antibodies were used: anti-K14 (1/1000, PRB-155P), anti-K10 (1/100, MMS-159S), anti-loricrin (1/500, PRB-145P) and anti-K6 (1/500, PRB-169P) (Covance); anti-Ki-67 (1/100, Abcam); anti-CD45 (1/100, BD Biosciences); anti-cleaved caspase-3 (1/250, 9661, Cell Signaling); anti-HOIP (custom-made, Thermo Fisher Scientific); and anti-HOIL-1[21]. Slides were incubated with the following secondary antibodies: Alexa Fluor 488 Goat anti-Rabbit IgG, 594 Goat

anti-Rabbit IgG (Invitrogen), or goat anti-rat horseradish peroxidase (HRP; Cambridge Bioscience) at room temperature for 1 h. Where an HRP-conjugated antibody was used, the TSA™ Plus Cyanine 3 System (Perkin Elmer) was applied according to the manufacturer's instructions. Sections were counterstained with 4,6-diamidino-2-phenylindole (DAPI; Roche). For HOIP and HOIL-1 staining, conventional immunohistochemistry (antibody dilution 1/100) was performed on BOND-III (Leica Microsystems) and BenchMark Ultra (Ventana-Roche Medical System) according to a protocol previously described[63]. For TUNEL staining, which was performed in combination with cleaved caspase-3 staining, the ApopTag Red In Situ Apoptosis Detection Kit (Merck Millipore) and streptavidin conjugate Cy2 (Jackson Immuno Research), respectively, were used according to the manufacturer's instructions. Sections were analyzed by fluorescent microscopy.

At least ten different images (×40) per slide were acquired. Quantification was performed by an experimenter who was blinded to the genotype of the samples by using the ImageJ Software on monochrome images as the percentage of cells positive for the specific staining in relation to the total number of cells (DAPI-positive) within the epidermis.

**Epidermal thickness quantification.** The epidermal thickness was measured in five different positions per microscopic field for at least ten different fields per mouse. Quantification was performed by an experimenter who was blinded to the genotype of the samples by using the ImageJ Software.

**Dermatitis scoring criteria.** Mice were assessed macroscopically based on two main clinical criteria. Each region of the body, comprising head, neck, back and flank, affected by lesions, was given a score of 1 and the sum of these provided information of how expanded the lesions were. The other criteria were the characteristics of the lesion: punctuated small crusts, coalescent crusts, and ulceration, with a score of 1–3, respectively. The sum of both criteria represented the total severity score of the lesions. Scoring was performed by two independent researchers.

**Isolation, culture and viability of PMKs.** PMKs were obtained from $Hoip^{E-KO}$ newborn pups, $Tnfr1^{KO};Hoip^{E-KO}$ and $Tnfr1^{KO};Hoil-1^{E-KO}$ adult tails according to established protocols[64]. Briefly, skin was incubated with 0.25% Trypsin in Hank's Balanced Salt Solution without calcium and magnesium (Stratech Scientific Ltd) overnight at 4 °C. On the following day, the dermis and epidermis were separated. Cell suspensions were cultured in Eagle's minimal essential medium (Lonza)

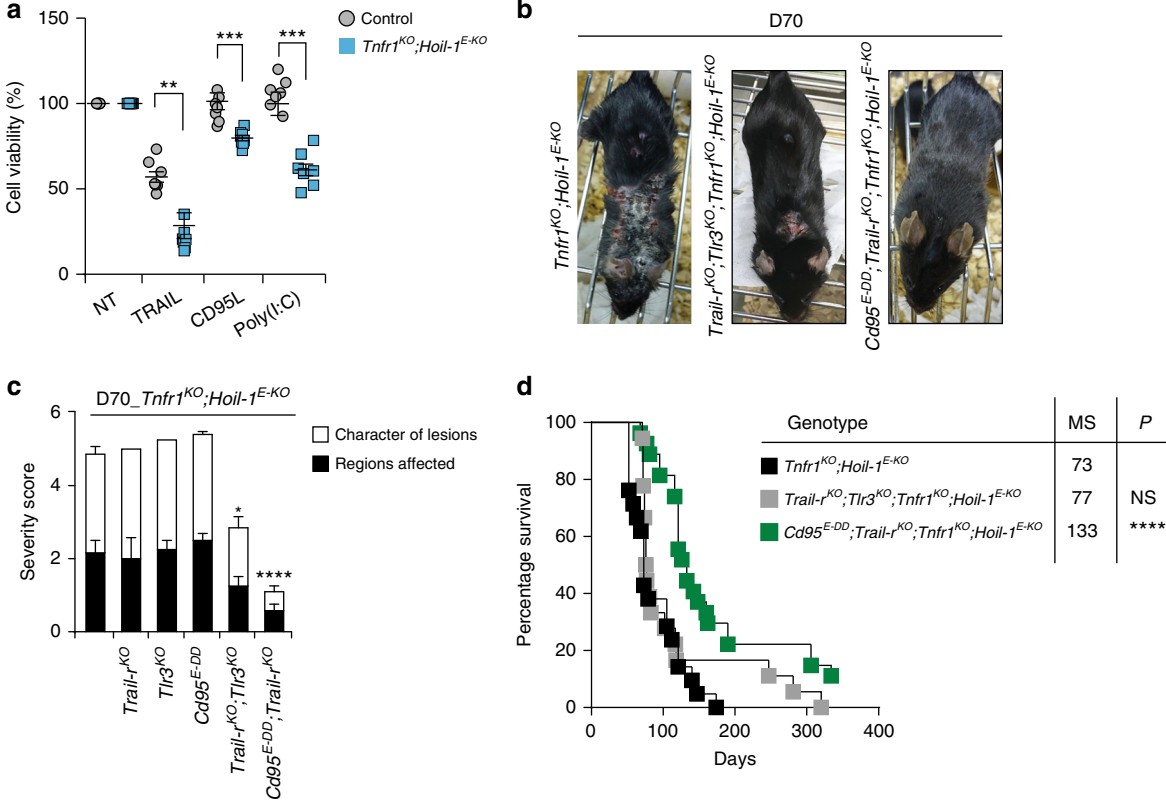

**Fig. 8** TNFR1-independent dermatitis is induced by CD95L- and TRAIL-driven cell death. **a** PMKs derived from adult mice of the indicated genotypes were left untreated or treated for 24 h with TRAIL, CD95L or Poly(I:C). Data are presented as mean values ± s.e.m. (n = 7 mice per genotype). **P ≤ 0.01, ***P ≤ 0.001. Control mice represent a pool of $Tnfr1^{KO};Hoil-1^{fl/fl};K14-Cre-$ and $Tnfr1^{KO};Hoil-1^{fl/wt};K14-Cre+$ mice. **b, c** Representative images (**b**) and severity score of dermatitis (**c**) of mice of the indicated genotypes assessed at D70. $Tnfr1^{KO};Hoil-1^{E-KO}$ (n = 6), $Trail-r^{KO};Tlr3^{KO};Tnfr1^{KO};Hoil-1^{E-KO}$ (n = 20) and $Cd95^{E-DD};Trail-r^{KO};Tnfr1^{KO};Hoil-1^{E-KO}$ (n = 19), *P ≤ 0.05, ****P ≤ 0.0001. **d** Kaplan–Meier survival curve of mice with the indicated genotypes. Comparisons between $Tnfr1^{KO};Hoil-1^{E-KO}$ mice with mice with the indicated genotypes were submitted for statistical analysis. MS: median survival. ****P ≤ 0.0001; NS: not significant. $Tnfr1^{KO};Hoil-1^{E-KO}$ (n = 21), $Trail-r^{KO};Tlr3^{KO};Tnfr1^{KO};Hoil-1^{E-KO}$ (n = 21) and $Cd95^{E-DD};Trail-r^{KO};Tnfr1^{KO};Hoil-1^{E-KO}$ (n = 28)

without calcium with 8% chelate foetal calf serum and penicillin–streptomycin (Sigma). PMKs were seeded in plates pre-coated with collagen I (Life technologies) for subsequent experiments. PMKs were cultured in medium supplemented with 20 μM Z-VAD-fmk (Abcam), 10 μM Necrostatin-1s (Cambridge Bioscience), 1 μM RIPK3 inhibitor (GSK2399872B) or 50 μg mL⁻¹ Etanercept (Enbrel®) (Pfizer and Pentaglobin from Biotest) for 4 days, with supplemented medium replaced every day. On the last day, cell viability was measured using the CellTiter-Glo Luminescent Cell Viability Assay Kit (Promega) following the manufacturer's instructions. Alternatively, PMKs were treated for 24 h with the following ligands as indicated: 50 ng mL⁻¹ mouse iz-TRAIL, 50 ng mL⁻¹ CD95L-Fc, or 100 μg mL⁻¹ Poly(I:C) (Invitrogen).

**Western blotting and immunoprecipitation (IP)**. Western blotting was performed as previously described[21]. Briefly, PMKs were lysed in IP-lysis buffer (30 mM Tris-HCl [pH 7.4], 120 mM NaCl, 2 mM EDTA, 2 mM KCl, 1% Triton X-100, EDTA-free proteinase inhibitor cocktail (Roche) and 1× phosphatase-inhibitor cocktail 2 (Sigma) at 4 °C for 20 min. Lysates were denatured with reducing sample buffer and dithiothreitol at 95 °C for 10 min. Proteins were separated by sodium dodecyl sulphate-polyacrylamide gel electrophoresis (NuPAGE) and analyzed by western blotting with antibodies (all primary antibodies were used at a 1/1000 dilution) against HOIP (custom-made, Thermo Fisher Scientific), HOIL-1[21], Sharpin (14626–1-AP, ProteinTech), actin (A1978, Sigma), tubulin (T9026, Sigma), FADD (Assay Design, AAM-121, RIPK1 (610459, BD), cleaved caspase-8 (9429, Cell signaling), MLKL (MABC604, Millipore), TNFR1 (ab19139, Abcam), phosphorylated IκBα (9246, Cell Signaling), IκBα (9242, Cell Signaling) and linear ubiquitin (MABS199, Millipore). Isolation of native TNFR1-SC and FADD IP were performed as previously described[26]. Briefly, PMKs were cultured in the presence of 20 μM Z-VAD-fmk (Abcam; broad spectrum caspase inhibitor) and, in the case of TNFR1-SC analysis, stimulated with 0.5 μg mL⁻¹ 3xFlag-2xStrep-TNF for the indicated times. Control cells were left untreated. Cellular lysates were subjected to anti-Flag IP using M2 beads (SIGMA; Schnelldorf, Germany) for 16 h. For FADD IP, lysates were incubated with anti-FADD

antibody (sc-5559, Santa Cruz) and protein G Sepharose Beads (GE Healthcare) at 4 °C for 4 h.

**Flow cytometry**. Cell suspensions obtained from skin samples were fluorescently labelled with Fixable Viability Dye eFluor® 780 (eBioscience). Samples were then stained with antibodies against the following cell surface markers: CD45-APC, CD45-AF700, CD3-PerCP/Cy5.5, CD4-FITC, CD8-PE/Cy7, GR1-FITC, GR1-PE/Cy7, F4/80-PE, F4/80-BV786, CD11b-Percp/Cy5.5 (Biolegend), and CD19-BV650 (Biolegend). Data were acquired with a LSRFORTESSA X-20 (BD) or Accuri (BD) with subsequent analysis using the FlowJo software.

**Statistics**. Data were analyzed with the GraphPad Prism 6 software (GraphPad Software) or Microsoft Excel. Data shown in graphs represent the mean values ± s.e.m., as indicated in the figure legends. Preliminary data sets were used to determine the group size necessary for adequate statistical power. Statistical analyses were performed by unpaired two-tailed Student's t test. For multiple grouped comparisons, two-way analysis of variance was applied. Statistical significance in survival curves was determined using a log-rank test. A P value of >0.05 was considered not significant (NS), whereas P ≤ 0.05 was indicated with one asterisk (*), P ≤ 0.01 with double asterisks (**), P ≤ 0.001 with triple asterisks (***) and P ≤ 0.0001 with four asterisks (****). In all cases, comparisons were made between the indicated knockout (KO) mice and the respective littermate controls.

## Data availability

All data are available from the authors upon request. Additional information on this manuscript can be found in the Supplementary Information.

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

## Acknowledgements

We thank Dr Geert van Loo (VIB Inflammation Research Center, Gent, Belgium) for providing *K14Cre* mice; Dr Vishva Dixit and Dr Kim Newton (Genentech, Inc., South San Francisco, CA) for providing *Ripk3^{KO}* mice; Dr Razq Hakem (University of Toronto, Ontario) for *Casp8^{KO}* mice; Helena Draberova, Paul Levy and his staff at the Kathleen Lonsdale Building (UCL, London, UK) for technical assistance; and Lorraine Lawrence (NHLI, Imperial College, London, UK) for histology service. This work was funded by a Wellcome Trust Senior Investigator Award (096831/Z/11/Z) and an ERC Advanced grant (294880) awarded to H.W. and NHMRC grants (project 602516, program 1113133) awarded to J.S. and H.W. or P.B. and A.S., respectively; N.P. is supported by the Swiss National Science Foundation (P300P3_158509), P.B., J.S. and A.S. are supported by NHMRC fellowships (1042629 1107149 and 1020363, respectively). This work is dedicated to the memory of our beloved colleague and friend Dr Martin Leverkus.

## Author contributions

H.W. conceived the project. L.T. performed the majority of the experiments. L.T., N.P. and H.W. designed the research and co-wrote the manuscript. A.M. contributed with the immune cell analysis, S.K. generated *Mlkl^{KO}* mice. N.P assisted experimentally throughout. M.D. and P.D. assisted with in vitro experiments, T.H. helped with immune characterisation and A.S. performed the majority of the genotyping. A.A. and T.M. performed HOIP and HOIL-1 immunohistochemistry and helped interpreting these stainings. M.P. provided *Ripk1^{D138N}* mice, J.B. and P.J.G. provided RIPK3 inhibitor and GSK3540547A (GSK'547A), M.L. provided *K14CreER^{tam}* mice, technical advice and contributed with helpful discussion. J.S. and H.W. generated the *Hoip^{fl/fl}* mice. E.R., J.S., P.B., A.St. provided critical scientific insight, edited the manuscript and, together with H. W. and T.L.H., contributed to the generation of *Hoil-1^{fl/fl}* mice.

## Additional information

**Competing interests:** J.B. and P.J.G. are employees of GSK. The remaining authors declare no competing interests.

