## [Peer Review File · Nature Communications]

REVIEWERS' COMMENTS:

Reviewer #1 (Remarks to the Author):

Taraborrelli et al. have well-addressed concerns raised by two reviewers. The group has studied roles of LUBAC in the regulation of inflammation in this and previous studies. This study is follow-up of the previous ones however, with a new concept and a discovery on the TNF-independent cell death regulation via LUBAC components HOIP and HOIL-1L, this study would significantly contribute to the research field.

There is only one minor comment.

- For the bar graphs used in the figures especially the ones for the mouse experiments, it is better to plot dots on top of the bars so that the readers understand better the sample distributions/number.

Reviewer #2 (Remarks to the Author):

This is the modified version of the paper dealing with the changes the authors completed in their previous version, their study deals with lethal dermatitis and combined inhibition of TNF-, TRAIL- and CD95L-mediated cell death.

The authors completed several studies in order to close the gaps that were mentioned by the reviewers in their previous version. I believe that the paper in its current form should be published. It is worthy and comprehensive and above all interesting as well.

I do not have additional comments concerning the paper.

Remarks to all referees:

Firstly, we would like to thank the reviewers for deeming our submitted work interesting and worthy of publication in Nature Communications. Please find below our response to the point raised by referee #1.

Referee #1:

- For the bar graphs used in the figures especially the ones for the mouse experiments, it is better to plot dots on top of the bars so that the readers understand better the sample distributions/number.

We have changed all bar graphs for dot plots.